# Investigation on Cycling and Calendar Aging Processes of 3.4 Ah Lithium-Sulfur Pouch Cells

**Salimeh Gohari** [1,2,*], **Vaclav Knap** [2,3] and **Mohammad Reza Yaftian** [1]

1   Phase Equilibria Research Laboratory, Department of Chemistry, Faculty of Science, University of Zanjan, Zanjan 45371-38791, Iran; yaftian@znu.ac.ir
2   Department of Energy Technology, Aalborg University, 9000 Aalborg, Denmark; vkn@et.aau.dk
3   Faculty of Electrical Engineering, Czech Technical University in Prague, 166 27 Prague, Czech Republic
*   Correspondence: s.gohari@znu.ac.ir or s_goohary@yahoo.com

**Abstract:** Much attention has been paid to rechargeable lithium-sulfur batteries (Li–SBs) due to their high theoretical specific capacity, high theoretical energy density, and affordable cost. However, their rapid c fading capacity has been one of the key defects in their commercialization. It is believed that sulfuric cathode degradation is driven mainly by passivation of the cathode surface by $Li_2S$ at discharge, polysulfide shuttle (reducing the amount of active sulfur at the cathode, passivation of anode surface), and volume changes in the sulfuric cathode. These degradation mechanisms are significant during cycling, and the polysulfide shuttle is strongly present during storage at a high state-of-charge (SOC). Thus, storage at 50% SOC is used to evaluate the effect of the remaining degradation processes on the cell's performance. In this work, unlike most of the other previous observations that were performed at small-scale cells (coin cells), 3.4 Ah pouch Li–SBs were tested using cycling and calendar aging protocols, and their performance indicators were analyzed. As expected, the fade capacity of the cycling aging cells was greater than that of the calendar aging cells. Additionally, the measurements for the calendar aging cells indicate that, contrary to the expectation of stopping the solubility of long-chain polysulfides and not attending the shuttle effect, these phenomena occur continuously under open-circuit conditions.

**Keywords:** lithium-sulfur batteries; calendar aging; cycling aging



## 1. Introduction

Recent trends in the energy sector strive towards high-density energy storages. Among electrochemical energy storage options, lithium-sulfur batteries (Li–SBs) are an appropriate option, which meet this requirement. This is due to the high theoretical specific capacity (1672 Ah kg$^{-1}$) and energy density (2600 Wh kg$^{-1}$) of the sulfur cathode, which is approximately an order of magnitude greater than commercially available cathodes, such as $LiCoO_2$, $LiMn_2O_4$, and $LiFePO_4$ [1,2]. Furthermore, sulfur has the following advantages of being a naturally abundant element, which is inexpensive, ecologically friendly, and non-toxic [3–5]. Although Li–SB has great potential, there are still challenges that hinder its successful commercialization. The identified Li–SB practical issues are [6–10]:

1.  Solubility of the polysulphides $Li_xS_y$ ($Li_2S_x$, $4 \leq x \leq 8$) in the electrolyte → leading to loss of active material of the solid cathode and irreversible capacity fade [4,11,12];
2.  Low ionic and electronic conductivity of S, $Li_2S$, and intermediate Li–SB products → lead to enhancement of the internal resistance of the cell, which results in low energy efficiency [4,11];
3.  Expansion of sulfur (∼80% change upon lithiation) → causes pulverization of cathodic materials and loss of electrical connection with the conductive substrate [13];
4.  Shuttle effect → results in insulation of the Li anode surface by sulfides, gradually delays the fast access of Li, and leads to poor rate capability [4,11,12].

To solve these problems, it is necessary to understand their root causes. The Li–SB is assembled from various components and each of them contributes to and influences the final system. The individual components of the Li–SB are shown in Figure 1, together with their desired features and attributes [14–20];

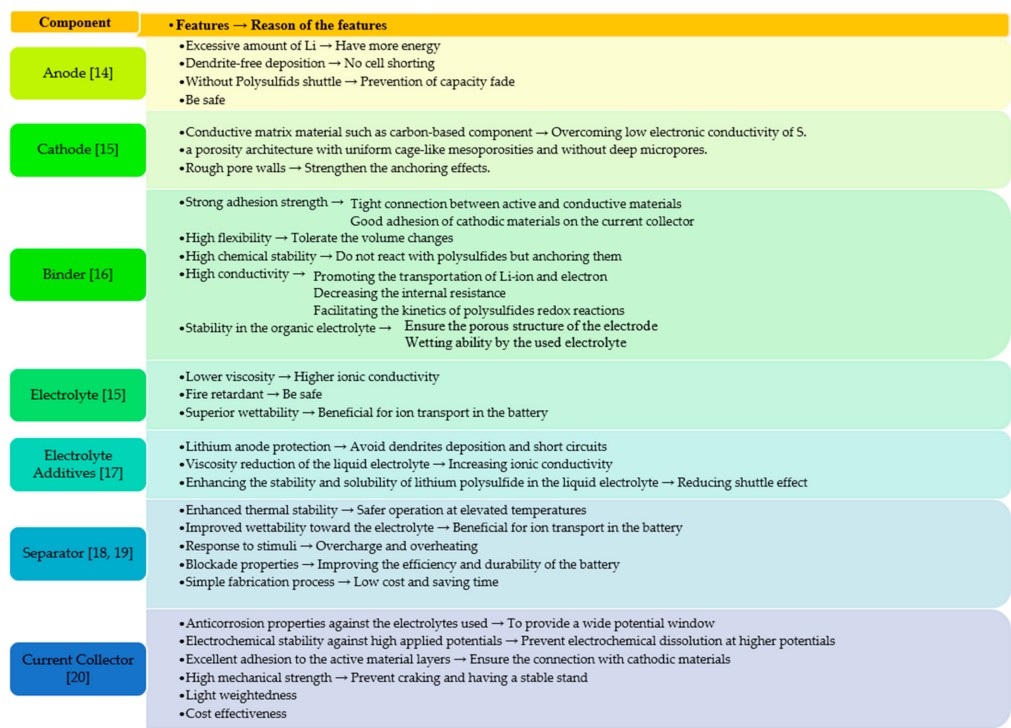

**Figure 1.** Battery components and their desired features.

5. The capacity loss can be linked to the degradation of the electrolyte components through side reactions with $Li^+$ resulting in less cyclable $Li^+$ in the electrodes by loss of active electrode material. The power losses consist of contact resistance in interphases between the materials, growth of resistive film on the active material, loss of active surface area, and impaired mass transport [21]. Generally, there are two main reasons that cause lithium- based batteries to deteriorate [21,22]:

6. Cyclable lithium fade: The performance of lithium-ion-based batteries depends on the transportation of lithium ions between the positive and negative electrodes of the battery. If the lithium ions lose their mobility during side reactions, then the battery performance will decrease;

7. Active materials fade: The active materials of a battery are those that participate in the electrochemical charge/discharge reaction. These include the two electrode materials of a cell and the electrolyte between them. If for any reason (such as material dissolution, particle isolation, electrode delamination, and structure degradation) these materials are subject to change, the battery capacity will decrease.

A comprehensive list of factors affecting the capacity loss and degradation of Li–SB is given in Figure 2. This shows connections between individual causes and the degradation mechanisms they lead to, resulting in the deterioration of Li–SB. In this list, the items indicated by solid lines include the proven items in the literature, and the dashed lines are based on the results of the analysis of lithium-ion batteries, and the validation of these items must be investigated and are yet to be proven [9,23–30].

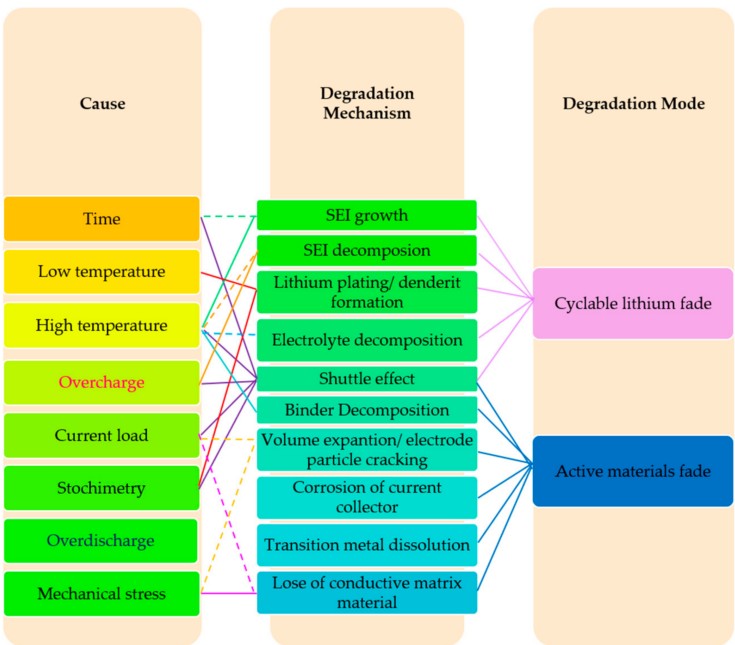

**Figure 2.** Network diagram of the causes and mechanisms of lithium- sulfur battery degradation. (Solid lines: indicates the proven items in the literature. Dashed lines: are based on the results of the analysis of lithium-ion batteries and are yet to be proven) [9,23–30].

A lot of research has been operated on lithium-ion batteries and the knowledge on this subject is provided in the literature with complete information available in reports presented by researchers. The performance of lithium-ion batteries is based on the insertion of lithium in the crystalline structure of the cathode (intercalation) all through the discharge process. However, this is different in the case of Li–SBs. The main electrochemical storage reactions of Li–SB can be described as follows, where sulfur is undergoing several steps of reduction in discharge, forming long and short chains of polysulfides in the process [31,32]:

Anodic half-reaction:

$$2Li \rightleftharpoons 2Li^+ + 2e^- \; 3.05 \; V$$

Cathodic half-reactions:

$$S_8 + 2Li^+ + 2e^- \rightleftharpoons Li_2S_8 \; \sim 2.4 \; V$$

$$3Li_2S_8 + 2Li^+ + 2e^- \rightleftharpoons 4Li_2S_6 \; > 2.3 \; V$$

$$Li_2S_6 + 2Li^+ + 2e^- \rightleftharpoons 2Li_2S_4 \; 2.1 - 2.3 \; V$$

$$Li_2S_4 + 2Li^+ + 2e^- \rightleftharpoons 2Li_2S_2 \; 1.9 - 2.1 \; V$$

$$Li_2S_2 + 2Li^+ + 2e^- \rightleftharpoons 2Li_2S \; < 1.9 \; V$$

Depending on the State of Charge (SOC), sulfur takes on different oxide states, from its elemental state $S_8$, via the intermediates $Li_2S_x$ ($1 \leq x \leq 8$). All these sulfur species react with lithium ions [33]. The occurrence of these different chains of chemical reactions makes it impossible to utilize traditional Li-ion batteries as learned concepts for Li–S [26,33]. More studies and research are required to improve Li−SB performance. Currently, the fundamental theory and practical knowledge of Li−SB is incomplete [26,33].

Parasitic processes can occur during shelf time due to the presence of electrons extracted from the lithium electrode, even without any external energy exchange, in addition to the parasitic ones that occur during cycling. Due to this, Li–SB useable capacity decreases because of reversible and irreversible reactions during storage [34]. The reversible portion of capacity fade is related to self-discharge and, after fully recharging, it can be recovered. In contrast, the irreversible fraction of capacity fading is divided into two parts: cycling

and calendar aging [34,35]. All the degradation phenomena that occur in a battery when the battery is charged or discharged can be attributed to cycling aging. The irreversible fraction of battery degradation depends on the electrodes and electrolytes with which the battery is assembled, and is independent of charge-discharge cycling, known as calendar aging [35,36].

Many of the previous observations were performed at small-scale cells (coin cells), while in this study, we investigated the degradation mechanisms of larger format (pre-commercial) Li–SB pouch cells. Processes of cycling and calendar aging were also studied. No external energy is applied under open-circuit conditions (referred to as calendar-life aging); therefore, the shuttle effect is not expected to occur (as one of the defections described in Figure 2). To confirm this expectation, a comparison was made between usage and storage time aged cells in order to determine the contribution of the shuttle effect in the degradation mechanism of cell performance. Selected performance indicators, i.e., capacity, coulombic efficiency, shuttle current, and resistance, were used to track and quantify the battery degradation. The shuttle current and $Li_2S$ precipitation were identified as the main drivers for degradation of the cycled cells, which exhibited more pronounced degradation in comparison to calendar aged cells. Moreover, it was observed that the shuttle current evolved during the aging process. The calendar case showed a linearly increasing trend, while for cycling, the trend first rapidly grew, followed by a decrease, which further underlines the complexity of Li–SBs.

## 2. Experimental Results

In this study, four pouch cells of 3.4 Ah Li–SB long-life type cells provided by OXIS Energy Ltd. were utilized for performing the experiments. The Digatron BTS 600 battery test station was used for performing tests and measurements. During the experiments, the cells were kept inside a thermal chamber at constant 30 °C, for both aging and performance evaluation. The remaining aging conditions of calendar aging cases (calTC-B1 and calTC-B2) were storage at 50% state-of-charge (SOC), and for cycling aging cases (cycTC-A1 and cycTC-A2), the cycling conditions consisted of full cycling (i.e., over the whole SOC region) with 0.1 C charging current and 0.2 C discharging current. A reference performance test (RPT) was performed every month for calendar aging cells, and every 20 cycles for cycling aging cells to monitor the selected performance indicators.

The RPT procedure has been described in detail in Ref. [37]. This procedure was utilized to characterize the cell at the beginning of life and during the periodical performance check-ups every month. It has six steps:

1.  A discharge step: obtaining the residual capacity of the cell after storage;
2.  A pre-condition cycle: for resetting 'a cumulative history' of the cell;
3.  A cycle to measure the cell's maximum charge and discharge capacity;
4.  Measuring the cell's resistance (at different SOC levels with 10% resolution, with 30 s long current pulses with amplitudes of 0.1, 0.2, and 0.5 C for charging and 0.2, 0.5, and 1 C for discharging);
5.  Measuring the shuttle current;
6.  Discharging the cell to the considered SOC level for storage.

The cycles were performed under the nominal condition, given in Table 1. The charging was additionally limited by a maximum charging time of 11 h.

For a straightforward comparison between cycling and calendar aging cases, the aging period of the cycling case is presented in terms of time, instead of cycles. The nominal cycle lasted approximately 15 h. The cycling period was 20 cycles. Thus, the periodical check at beginning of life lasted around 12.5 days.

**Table 1.** Thermal and electrochemical characteristics of the Li–SBs.

| Characteristics | Values |
|---|---|
| Nominal capacity (30 °C) | 3.4 Ah |
| Nominal charging current | 0.34 A (0.1 C-rate) |
| Nominal discharging current | 0.68 A (0.2 C-rate) |
| Nominal voltage | 2.05 V |
| Charge cutoff voltage | 2.45 V |
| Discharge cutoff voltage | 1.5 V |

## 3. Results and Discussion

### 3.1. Capacity Fading Evaluation Trend

The evolution of the charging and discharging capacity of Li–SBs during aging is presented in Figure 3a,b, respectively. For the cycling case, the charge capacity, after an initial dip, increases gradually over the first 61 days, and then it plateaus as the charging period becomes prolonged due to the strong presence of polysulfide shuttle, preventing the cells from reaching the cutoff voltage limits; instead, the charging is limited by time. The charging capacity remains constant until day 178, when it starts to decrease over time. The calendar aging exhibits different behavior, whereas the charging capacity is almost constant and shows negligible changes during the aging period. When observing the discharge capacity, it can be seen that it is decoupled from the charging capacity. Both aging cases, cycling and calendar, show a monotonic downward trend for the discharge capacity. In the case of calendar aging, the discharge capacity decrease has a linear character, and it is slower than in the cycling case. The discharge capacity during cycling has a nonlinear trend, where it first exhibits a fast decline, followed by a steady, slow decrease until a 'knee' is reached, and a rapid fall in capacity occurs.

Since the capacity is measured by the total flow of $Li^+$ ions or electrons to/from the electrode and usually, the total capacity is limited by the cathode, coulombic efficiency (CE) is the more appropriate index for estimating the cycling life of rechargeable batteries [38]. By definition, CE is the ratio between discharge capacity to charge capacity [38]. In the case of Li–SBs, electrons are readily available during charging on the anode side; therefore, long-chain polysulfides could easily be reduced and the shuttle effect occurs. These short-chain polysulfides migrate back to the cathode electrode, where they could be oxidized again. The electron supply for these electrochemical reactions does not contribute to the externally measured current, and this is the main reason for low CE [39]. The evolution of CE during aging is presented in Figure 3c. For calendar aging, the charged cycle capacity generally remains constant, while the discharged cycle capacity which decreases gently. Accordingly, their ratio demonstrates a lenient downward slope.

With regards to cycling aging, by increasing the difference between discharge and charge capacity over 61 days the CE decreases. In the period of 61 to 178 days of the cycling aging cases, CE does not show a sharp downward trend, and it has a very gentle decreasing trend instead. In this period, the charged capacity remains constant, because the cutoff voltage was not reached, and the charging was stopped by a time limit due to the strong presence of the polysulfide shuttle at the end of charging. On the other hand, for the discharged capacity, the polysulfide shuttle has only a minimal effect, as the cells do not stay at very high SOCs for a prolonged time. Hence, fewer differences between charge and discharge capacity lead to the appearance of this gentle decreasing trend. After 178 days for the cycling aging cases, CE begins to increase. Over time, the amount of short-chain polysulfides at the cathode surface increases and prevents the long-chain polysulfides from dissolving, thus reducing the concentration of long-chain polysulfides that could migrate to the anode via the electrolyte to generate a shuttle current. As a result, the charge capacity is less affected by the shuttle current, its ratio to discharge capacity increases, and the CE gains are found [39].

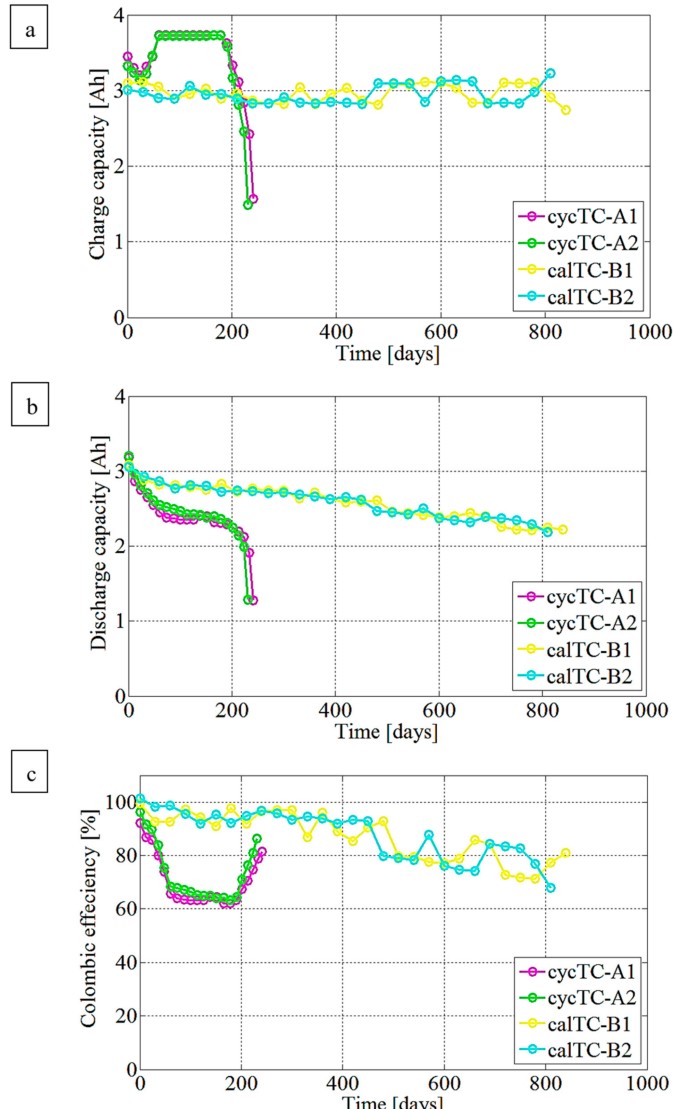

**Figure 3.** Cycling performance profiles include (**a**) charge capacity, (**b**) discharge capacity, (**c**) coulombic efficiency.

　　Figure 4a shows the shuttle current, which confirms the observed trend for CE. Shuttle current profiles indicate a gradual increase in the concentration of polysulfides in the electrolyte, although, for cycling aging, the concentration of soluble polysulfides gradually becomes less fenced in, due to the short-chain polysulfides at the positive electrode surface. Thus, this trend in the shuttle current over time becomes an indicator of the ensuing irreversible loss of cathodic active material and capacity fade [39].

　　For a deeper insight, a cell's total capacity can be split into a capacity of high- and low-voltage plateaus, illustrated in Figure 4b,c for cycling and calendar aging, respectively. The available discharge capacity of the high-voltage plateau is generally related to the severity of the shuttle current, whereas, in the low-voltage plateau, this is attributed to the electrode structure cracking due to successive precipitates of $Li_2S$ on both electrode surfaces. On the cathode side, electrochemical reduction caused the formation of $Li_2S$, and on the anode side, $Li_2S$ was formed by the chemical reduction in polysulfides that diffuse from the cathode [30].

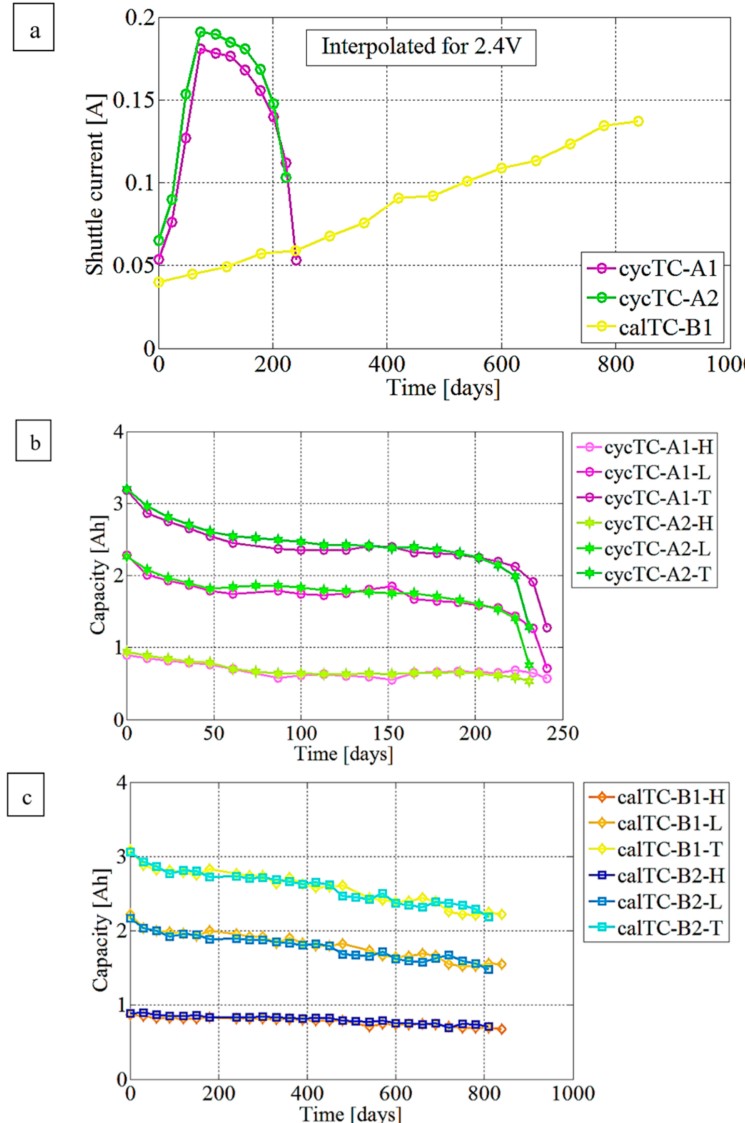

**Figure 4.** (**a**) Shuttle current (interpolated for 2.4 V) for cycling and calendar aging, and the high and low voltage plateau profiles for (**b**) cycling aging and (**c**) calendar aging of Li–SBs.

### 3.2. Resistance Evaluation Trend

Cell resistance impacts the extractable/available cell capacity because cell resistance is correlated to cell capacity [40,41]. Higher resistance leads a cell to reach the discharge cut-off voltage limit earlier and makes the available discharge capacity lower [40]. Moreover, the relationship between resistance and current flow indicates that the charge transfer overpotential is one of the determinative parameters in electrochemical kinetics as it expresses the resistance reflection to the electrode surface reaction [42].

It was reported that at high C-rates the resistance increases due to the increased passivation of the cathode surface by $Li_2S$ (higher C-rates produce a high nuclei density, and thus many small crystallite morphologies are created; at low C-rates, fewer but bigger precipitates are produced) [43]. However, in the investigated cells, the computed resistance does not vary for various C-rate pulses (Figure 5a,b: the evolution trend of Li–SBs resistance of cycling and calendar aging in different C-rates at various SOCs). This is because of the sufficiently fast kinetics of the supplied lithium- ions participating in the reactions for those C-rates. Furthermore, for the C-rates used in this study, all dissolved species have enough rapid mobility in the liquid electrolyte to create a negligible difference in overpotential among the reactions operated during these currents [39]. However, in the case of the 10%

SOC, the curves are slightly different and do not overlap. On this subject, it was previously proven that in SOC levels above 75%, the elemental sulfur is reduced to polysulfides and forms the long-chain polysulfides ($Li_2S_x$, $4 \leq x \leq 8$), which are soluble in the organic electrolyte) [44–46]. However, for SOCs less than 75%, the long-chain polysulfides undergo further reductions into short-chain polysulfide form ($Li_2S$ and $Li_2S_2$, which are insoluble in the organic electrolyte). Hence, the amount of $Li_2S$ produced in 10% SOC is higher, and its effect causes transport limitations at higher discharging C-rates, resulting in a higher resistance compared to the charging process. The formation of undissolved $Li_2S$ is increased by cycling [35]. Thus, the increased resistance at a low SOC is stronger in the cycling aging case, compared to the calendar aging.

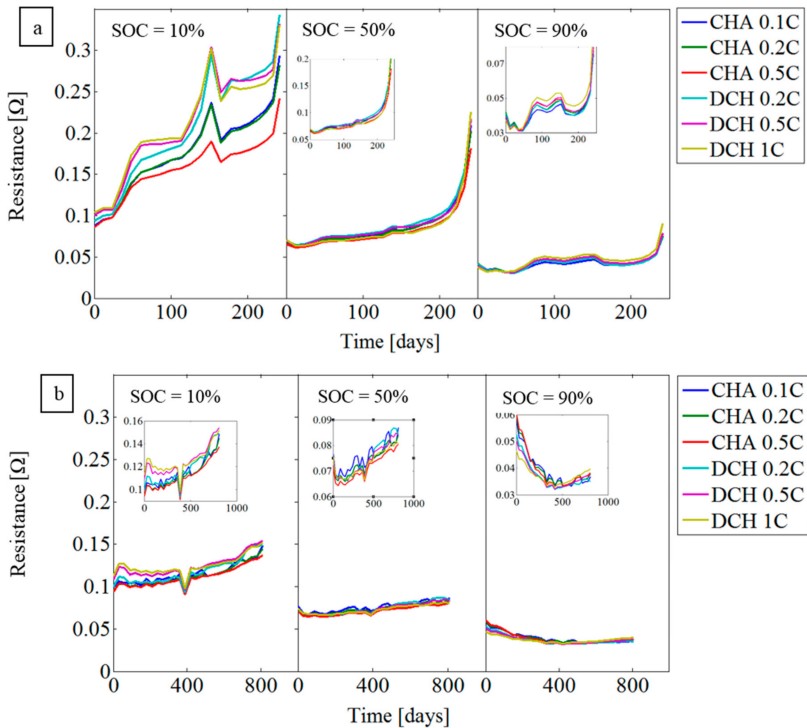

**Figure 5.** Internal Li–SBs resistance of (**a**) cycling (cycTC-A1) and (**b**) calendar (calTC-B1) aging measured at various C-rates and SOCs for the 30th second of discharge pulses.

The patterns of the evolution trend of Li–SBs internal resistance versus aging time for various SOCs for cycling and calendar aging in the first second and 30-s after applying of the pulse, as well as their differences (30-s minus the first second) are depicted in Figures 6 and 7, respectively. The internal resistance for high SOCs (90%) has a lower value than for low SOCs (10%). These results are in agreement with Refs. [44,47]. Chien and et al. [48] have justified these observations, reporting that the crystalline sulfur species formation and disappearance were correlated directly with the internal resistance trend and diffusion resistance coefficient. Insoluble forms of crystalline sulfur during the discharge process terminate the porous carbon matrix by pore blockage. Correspondingly, the utilization of sulfur is limited, and consequently the specific capacity of the sulfur/carbon composite electrode becomes low [48]. Additionally, it is reported that the cathode surface is passivated by $Li_2S$. This passive layer creates limitations against the passage of lithium ions [49]. Another reason for this is that, at low SOCs, the main active materials are short-chain polysulfides and sulfur. As we know, these sulfur species have low electronic conductivity, causing a high level of resistance against reactions [43,50]. Thus, the high amount of short-chain polysulfides at low SOCs is the cause of the increasing internal resistance.

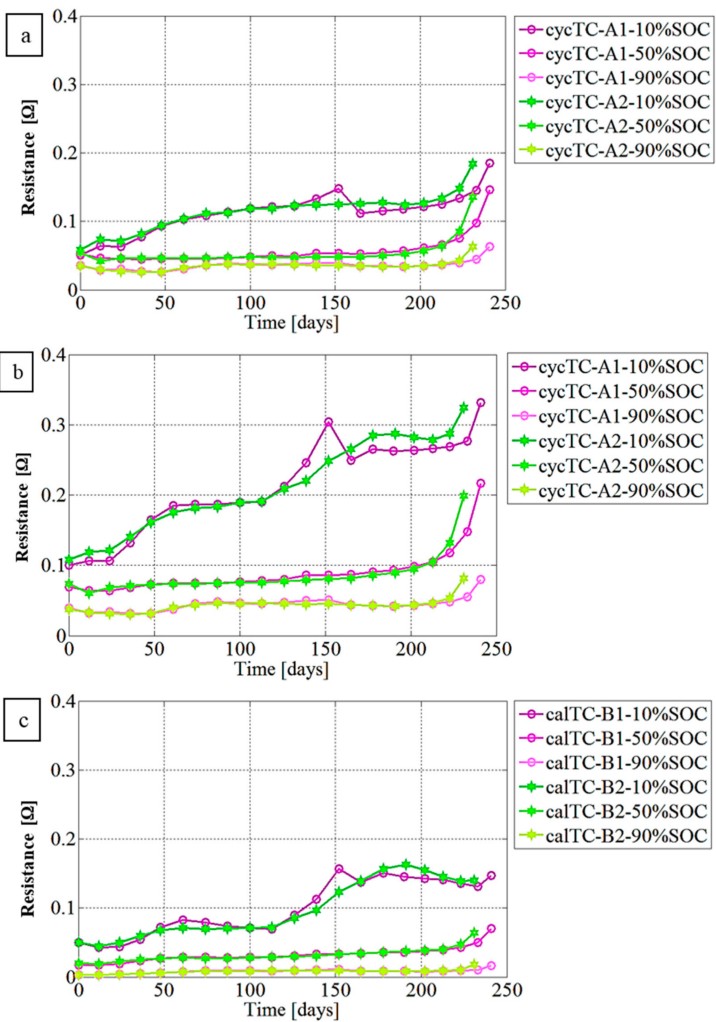

**Figure 6.** Internal Li–SBs resistance of discharge cycling aging for (**a**) 1 s, (**b**) 30 s (**c**) (30−1) s for 10%, 50%, at 90% SOCs and C-rate of 0.5.

At each discharge cycle, three forms of sulfur exist: The first fraction is elemental sulfur (i.e., solid sulfur, not dissolved), the second fraction is $Li_2S$, and the third fraction is dissolved polysulfides. The concentration of these three factions, in addition to the SOC and C-rate, depends on time. The concentration of the third fraction, insoluble polysulfides forms which enhance internal resistance, gradually increases, and at the end of the discharge, some of the elemental sulfur remains and does not convert to dissolved forms [39]. Hence, the internal resistance for the first second is less than that for the 30th second. The reason for the sudden increase in internal resistance in cyclic aging on day 213 onwards is clogging of the sulfur/carbon composite porous matrix, and the surface of the cathode and its partial passivation [39]. Another explanation is interpreted as follows: at the initial time of a cycle, the ionic conductivity of the used electrolyte is sufficient for conducting electrochemical reactions [51]. By the dissolution of long-chain polysulfides, the viscosity of the electrolyte increases and leads to the reduction in ionic conductivity in conformity with the Stokes−Einstein relationship [51–53]. In addition, precipitation of short-chain polysulfides tends to fade electrical contact [51,54]. The rate of change of resistance is greater in the first second (Figures 6a and 7a) than the rest of the time until the 30th second (Figures 6c and 7c); this is related to the reduction in the kinetics of the electrochemical reaction after the dissolution−precipitation of polysulfides in comparison to the relatively ideal conditions prevailing in the first second of a cycle [51].

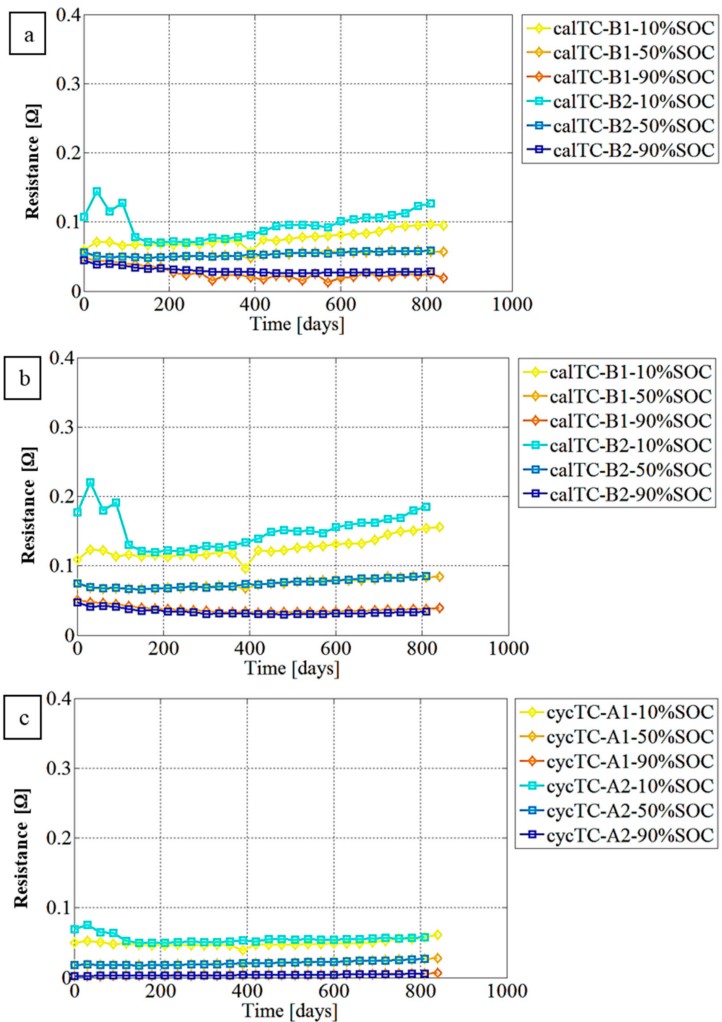

**Figure 7.** Internal Li–SBs resistance of discharge calendar aging (**a**) 1 s, (**b**) 30 s (**c**) (30−1) s for 10%, 50%, at 90% SOCs and C-rate of 0.5.

By comparing Figure 6a,b and Figure 7a,b, it can be seen that the internal resistance trends for calendar and cycle aging are different. In cyclic aging, the trend of internal resistance increases for 241 days and jumps suddenly in the later days, but in calendar aging at 840 days, its trend is linear with a gentle slope. These observations show that even under storage state, parasitic electrochemical reactions take place spontaneously. In the case of cyclic aging, the reason its internal resistance increases earlier than in the case of calendar aging is that as the Li–SBs are cycled, the parasitic reaction rate will increase from cycle to cycle, and the cathodic porous carbon matrix and surface gradually are blocked and terminated [30,39,48,55].

## 4. Conclusions

The cycling and calendar aging of Li–SBs was investigated in this work via measurements of resistance, capacity, coulombic efficiency, and shuttle current. The study focused on a pre-commercial 3.4 Ah Li–SB pouch cell, unlike most of the other previous observations that were done on small-scale cells (coin cells). The process of battery degradation in cyclic aging is faster than calendar aging because the fraction of cathodic active material decreases from cycle to cycle. Furthermore, the analysis shows that under open-circuit conditions in storage time, the solubility of long-chain polysulfides and the shuttle effect occur continuously, contrary to the initial expectation.

Capacity fading increases over time. Gradual capacity fading was observed for the calendar aging with an increasing trend and a light slope versus a sharp increase followed

by a sharp decreasing trend for cycling aging. Although the charging process is incremental, it is restricted by time and voltage limitation. Therefore, in the plots, an upward, constant and decreasing trend was observed. It may be concluded that the shuttle effect is the main cause of degradation, and the observed capacity fade trend is consistent with the shuttle current.

SOC has an influence on internal resistance: at low SOCs, the amount of poor electronic conductivity of sulfuric species is high, blocking the porous matrix and passivating the cathode, resulting in an increase in internal resistance. Since the effect of higher resistance is less discharge capacity at a specific C-rate, resistance changes are a good performance indicator for battery aging. The effect of C-rates, which was applied in this study, on internal resistance, was not observed because there were no limitations for the speed of reactions and negligible differences among their overpotentials.

**Author Contributions:** Conceptualization, S.G. and V.K.; methodology, V.K.; software, S.G.; validation, S.G., V.K and M.R.Y.; formal analysis, S.G.; investigation, S.G.; resources, S.G.; data curation, S.G. and V.K.; writing—original draft preparation, S.G.; writing—review and editing, V.K.; visualization, S.G.; supervision, V.K. and M.R.Y.; project administration, V.K.; funding acquisition, V.K. All authors have read and agreed to the published version of the manuscript.

**Funding:** This research received no external funding.

**Institutional Review Board Statement:** Not applicable.

**Informed Consent Statement:** Not applicable.

**Data Availability Statement:** The data is available upon request from the corresponding author.

**Acknowledgments:** The authors would like to thank OXIS Energy (Abingdon, Oxfordshire, UK) for supplying the Lithium-Sulfur battery cells.

**Conflicts of Interest:** The authors declare no conflict of interest.

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
