# Peer review of "Investigation on Cycling and Calendar Aging Processes of 3.4 Ah Lithium-Sulfur Pouch Cells"

_sustainability, doi:10.3390/su13169473_

Round 1

Reviewer 1 Report

This paper reports the cycling and calendar aging processes of commercial Li-S pouch cells by measuring resistance, capacity, coulombic efficiency, and shuttle current. After carefully read the manuscript, some advice is given for the authors because of the following reasons:

  1. There are some errors and grammar mistakes in this form paper, and the authors should check it carefully and must be improved the quality of writing.
  2. Abbreviations should be the same throughout the paper. For example, it is referred to as Li-S in some places and as Li-SB in other places. Either Li-S or Li-SB should be selected.
  3. The novelty of this study should be expressed precisely at the end of the introduction section.
  4. In order to understand more about the state of the charge mechanism, the potentials level (Voltage) of phase transformation reactions should be given on the 4th page. This information would help to understand the "3.2. Resistance evaluation trend" section.

Author Response

Dear reviewer,

On behalf of my coauthors, I would like to thank you for considering our manuscript for possible publication in the "Sustainability journal". 

Reviewer 2 Report

The work of Salimeh Gohari and coworkers investigates the effects of cycling and calendar aging processes  on the electrochemical performance of 3.4 Ah Lithium-Sulfur Pouch Cells, by measuring resistance, capacity, coulombic efficiency, and shuttle current. The study seems more interesting and important for the commercialization of Li-S batteries since it focuses on a pre-commercial (factory-scale) 3.4 Ah Li-S pouch cell, as compared to the previous studies carried out at small/laboratory-scale cells (coin cells).

One of the drawbacks of this article, the authors don't bring a solution or any suggestion to overcome all challenges mentioned in the conclusion part; such as (i) why the battery degradation during the cyclic aging is faster than calendar aging?, (i) why the solubility of long-chain polysulfides and the shuttle effect continues to occur even under OCV condition in storage time, and (iii) the reason for why capacity fade increases over time, or any suggestion/remedy for this challenge?, etc. These questions should be answered or addressed clearly, especially in the conclusion. 

The other analysis provided by the authors are fully in line with the presented data, and provide interesting insights for the battery community. The article is not vague, well-written and provide a detailed experimental section.

I therefore suggest to publish this manuscript as is, with the only exception for the figures: so my other suggestions for the authors are;

1) to increase the font of any text in the figures (it is really hard to read them, especially Fig 2 and Fig 4), as well as their overall quality (at least in the pdf version for the reviewers, some figures tend to be "pixelated").

2) The article should be carefully reviewed/checked one more time in terms of typo mistakes; as seen in the last sentence of the conclusion part; as follows: ...... cause of (it should be "because of") no limitation for speed of reactions and negligible differences among their overpotentials.

Author Response

(The authors gave the same response as above.)

Reviewer 3 Report

I believe that the paper entitled "Investigation on Cycling and Calendar Aging processes of 3.4 Ah Lithium-Sulfur Pouch Cells" is suitable for publication in Sustainability after the authors consider the following points:

  1. Could you please include information related with the small scale pouch cells? Are there any restrictions using the particular protocols?
  2. Are there any restrictions on the measurement of large scale cells up to the present? Did you consider any assumptions related with the cells characteristics (i.e. dimensions, distances) prior to aging analysis? Please comment on that.

Author Response

(The authors gave the same response as above.)
